# VidSplice: Towards Coherent Video Inpainting via Explicit Spaced Frame Guidance

## Abstract

Recent video inpainting methods often employ image-to-video (I2V) priors to model temporal consistency across masked frames. While effective in moderate cases, these methods struggle under severe content degradation and tend to overlook spatiotemporal stability, resulting in insufficient control over the latter parts of the video. To address these limitations, we decouple video inpainting into two sub-tasks: multi-frame consistent image inpainting and masked area motion propagation. We propose VidSplice, a novel framework that introduces spaced-frame priors to guide the inpainting process with spatiotemporal cues. To enhance spatial coherence, we design a CoSpliced Module to perform first-frame propagation strategy that diffuses the initial frame content into subsequent reference frames through a splicing mechanism. Additionally, we introduce a delicate context controller module that encodes coherent priors after frame duplication and injects the spliced video into the I2V generative backbone, effectively constraining content distortion during generation. Extensive evaluations demonstrate that VidSplice achieves competitive performance across diverse video inpainting scenarios. Moreover, its design significantly improves both foreground alignment and motion stability, outperforming existing approaches.

## 1 Introduction

Video inpainting, the task of generating or removing specific elements within a video based on given masks, sits at the intersection of creativity and perception, representing one of the most challenging problems in computer vision. Its inherent difficulty stems from the dual requirement of spatial fidelity and temporal coherence: each inpainted frame must be visually indistinguishable from its surroundings, while the sequence as a whole must evolve naturally over time, free from temporal artifacts, distortion, or flickering. Recent advances in Diffusion Transformers (DiTs) Ho et al. (2020); Peebles & Xie (2023); Rombach et al. (2022) and video generative priors Hong et al. (2022); Yang et al. (2024); Wang et al. (2025) have brought unprecedented generative quality, offering a promising avenue to tackle temporal consistency with remarkable fidelity.

Traditional approaches, relying predominantly on 3D CNNs Chang et al. (2019a); Hu et al. (2020); Wang et al. (2019), often struggle when capturing long-range temporal dynamics, producing temporally unstable results. Subsequent methods enhanced temporal modeling via optical flow guidance Zhou et al. (2023) or by leveraging pretrained image-to-video (I2V) generative models Zi et al. (2025); Wan et al. (2024); Li et al. (2025); Gu et al. (2024); Bian et al. (2025); Jiang et al. (2025). For instance, COCOCO Zi et al. (2025) injects motion-aware modules into T2I backbones for text-guided video inpainting, while DiffuEraser Li et al. (2025) fuses BrushNet Ju et al. (2024) with diffusion models to achieve consistency over time. Complementarily, video editing methods Meng et al. (2021); Ceylan et al. (2023); Geyer et al. (2023) employ attention or structure guidance to manipulate appearance, but they fall short in reconstructing missing semantics, particularly when identity alignment and temporal coherence are critical.

In some real-world scenarios, video inpainting necessitates processing videos in chunks. Image-to-Video based models are inherently well-suited to address this challenge by employing a sliding window, where the last frame of the preceding chunk serves as the initial frame for the subsequent chunk to maintain inter-chunk consistency. Despite these successes, existing methods often rely solely on a single inpainted frame Rombach et al. (2022); Labs (2024) as reference and delegate

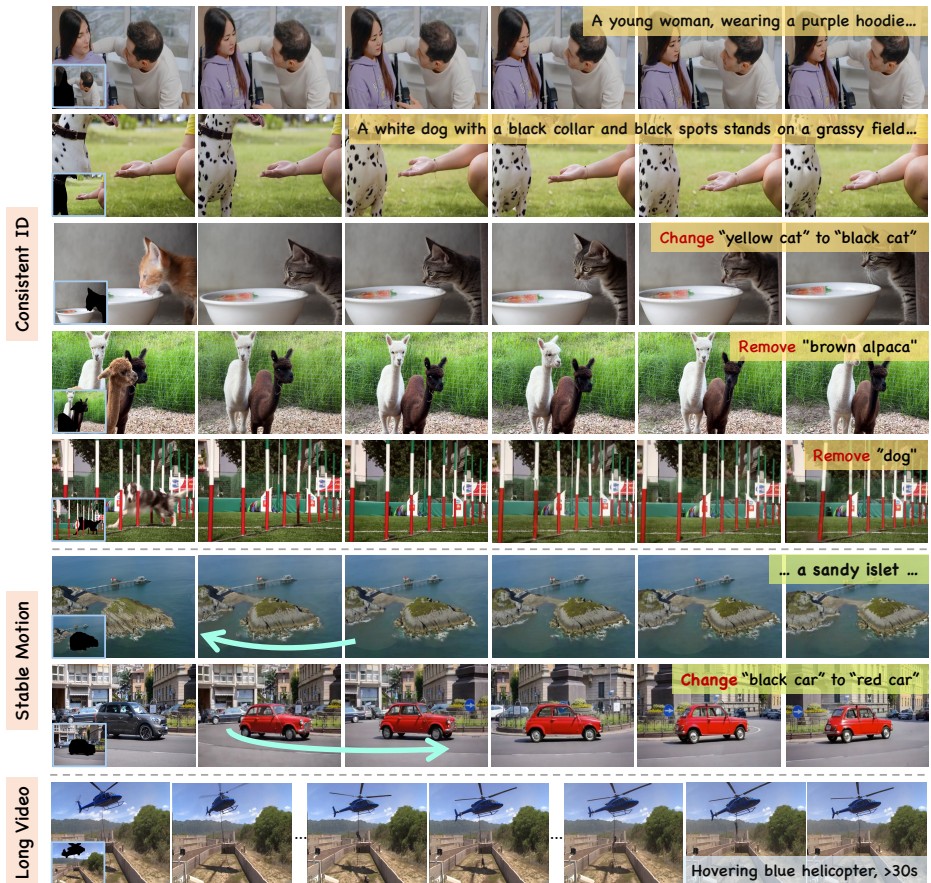

Figure 1: The inpainting results of our VidSplice. VidSplice can preserve consistent ID across inpainted / edited videos, achieve stable motion generation and viewpoint transformation.

whole motion learning to a pretrained image-to-video model Guo et al. (2023); Hong et al. (2022); Yang et al. (2024). This implicit handling of masked area motion can lead to severe artifacts when motion estimation is suboptimal, especially in the case of dynamic scenes, resulting in incomplete regions, content distortion, or a failure to maintain spatiotemporal consistency, as illustrated in Fig. 4.

To address these challenges, we propose VidSplice, a novel video inpainting framework that systematically decouples the task into two complementary subproblems: maintaining per-frame spatial coherence and ensuring cross-frame temporal consistency. Specifically, we introduce the CoSpliced module for multi-frame consistent image inpainting, which ensures temporal coherence across the video while preserving spatial consistency within each frame. To further augment temporal stability, we propose a first-frame propagation mechanism. This mechanism diffuses the content from the initial inpainted frame to subsequent sampled frames through a learnable splicing operation. Complementing this, we introduce masked area motion propagation to preserve short-range temporal and motion dynamics. Central to the VidSplice framework is the strategic use of spaced-frame guidance. By leveraging sparsely sampled frames, our model is supplied with structured temporal cues that guide the inpainting process. Moreover, we incorporate a context controller module that encodes and injects contextual video information, capturing coarse camera and motion dynamics while constraining the inpainting process across the entire sequence. Qualitative results on real-world videos demonstrate that VidSplice preserves consistent identities across inpainted and edited sequences, achieves stable motion generation and viewpoint transformations, and generalizes well to a varity of scenarios, as illustrated in Fig. 1.

Our contributions can be summarized as follows:

- We propose a novel framework VidSplice for video inpainting and editing, which successfully maintains spatiotemporal consistency and generalizes well to various video scenarios.

- We propose the CoSplice Module, an innovative component that generates temporally consistent context video streams using priors from an image inpainting model.

- We develop a context controller that effectively extracts and injects contextual motion features from context video into a diffusion process of whole video inpainting.

- Extensive experiments demonstrate that VidSplice achieves state-of-the-art performance in different benchmarks. It exhibits strong foreground alignment and motion stability.

## 2 RELATED WORKS

**Video Inpainting.** Video inpainting aims to model the spatiotemporal dependencies within masked regions and has seen significant progress in recent years. Early approaches Chang et al. (2019a); Hu et al. (2020); Wang et al. (2019); Chang et al. (2019b) typically employed 3D CNNs to capture local spatiotemporal patterns. However, they always struggled with long-range temporal propagation. More recent methods Zi et al. (2025); Wan et al. (2024); Li et al. (2025); Gu et al. (2024); Bian et al. (2025); Cho et al. (2025); Jiang et al. (2025) have shifted toward leveraging pretrained video generative models, which enable temporally coherent inpainting by better capturing motion dynamics and long-range temporal consistency. Among them, the most similar work to ours is VideoPainter Bian et al. (2025) and VACE Jiang et al. (2025), both leverage video diffusion models for temporal modeling and VideoPainter use an ID adapter for ID preservation. However,VideoPainter struggles to preserve consistent ID in some inpainting scenarios and introduces noticeable motion artifacts due to error accumulation of ID clips. In contrast, our VidSplice employs CoSpliced to generate consistent frames for direct guidance and enjoys persistent ID alignment.

**Video Editing.** Video editing methods Ku et al. (2024); Cheng et al. (2023); Esser et al. (2023); Wang et al. (2023); Jiang et al. (2025) primarily focus on modifying the properties of existing elements, such as their color, style, or texture. Some works Cheng et al. (2023); Singer et al. (2024) perform video editing through text-based control, while others incorporate structured data Esser et al. (2023); Wang et al. (2023); Liang et al. (2024a), such as depth, to provide additional spatial constraints during the generation process. Due to the lack of annotated datasets for fine-grained editing training, many approaches Ceylan et al. (2023); Geyer et al. (2023); Kara et al. (2024); Khachatryan et al. (2023) resorted to training-free strategies. Pix2Video Ceylan et al. (2023) used structure-guided diffusion model to edit first frame and propagate to other frames by attention in latent space. FairyWu et al. (2024) and TokenFlowGeyer et al. (2023) adopt a similar strategy of performing edits on selected frames of a video sequence. However, unlike our approach, their methods manipulate attention during the denoising steps. Moreover, their tasks are formulated as video-to-video translation, which are well-suited for stylization and appearance transfer. In contrast, our method starts from a masked video and focuses on generating ID-aligned content. The explicit use of masks in our framework provides more flexibility for editing.

**Controllable Image Generation and Inpainting.** Recent advances in controllable image synthesis Ruiz et al. (2022); Zhang & Agrawala (2023); Mou et al. (2023) and image inpainting Li et al. (2022); Suvorov et al. (2022); Dong et al. (2022) have greatly enhanced the capability of generative models in visual tasks. ControlNet Zhang & Agrawala (2023) and T2I-Adapter Mou et al. (2023) introduce trainable adapters Houlsby et al. (2019) that inject visual conditions into pretrained backbones for controllable generation. Reference-guided inpainting methods Zhou et al. (2021); Zhao et al. (2022b;a); Cao et al. (2024); Tang et al. (2024); Oh et al. (2019) leverage additional reference images from different viewpoints to inform the completion of target image. Among them, Leftrefill Cao et al. (2024) explores the use of prompt tuning for improving identity consistency in reference inpainting. While effective in maintaining visual fidelity across views, the generation of Leftrefill can not controlled by language prompts and also fails to generalize to the video domain of dynamic scene.

## 3 METHODOLOGY

**Problem Definition.** Given a video sequence $V = \left\{ V_t \in \mathbb{R}^{H \times W \times 3} \right\}_{t=1}^{T}$ of $T$ frames, along with corresponding binary mask for each frame $\mathbf{M} = \left\{ M_t \in \mathbb{R}^{H \times W \times 1} \right\}_{t=1}^{T}$, the objective of video inpainting is to generate or remove specific elements with the masked video $\widetilde{V} =$

Figure 2: The overall pipeline of VidSplice. As shown in the figure, VidSplice consists of three main components: the CoSpliced module, the context controller, and the video diffusion transformer. The CoSpliced module generates ID-aligned anchor frames, while the context controller extracts the upstream-constructed context video and injects it into the video diffusion transformer to guide temporal motion modeling.

$\left\{ V_t \odot (1 - M_t) \in \mathbb{R}^{H \times W \times 3} \right\}_{t=1}^T$ as input, guided by a textual or visual prompt $P$. The inpainted region is excepted to be spatially seamless and temporally consistent throughout the entire sequence.

**Overview.** Fig. 2 provides a overview of VidSplice. Specifically, VidSplice takes a masked video $\widetilde{V}$ and a textual prompt $P$ as inputs. The masked video is first processed by the carefully designed CoSplice Module (Sec. 3.1), which produces two distinct spliced video streams: the I2V-masked video and the context video. Each stream is constructed using priors derived from an image inpainting model and incorporates identity alignment across frames to ensure instance-level temporal consistency. Then 3D VAE encoder is employed to produce $8\times$ spatiotemporal downsampled latent features for these two video streams. We employ a context controller to extract contextual feature from the context video, serving as constraint of appearance preservation (Sec. 3.2). Finally, a diffusion transformer accepts the I2V-masked video as input, guided by the contextual feature, and produce temporally coherent inpainting results (Sec. 3.3). So, VidSplice's design offers key advantages for video inpainting:

- Reference-guided context image inpainting can successfully preserve the identity consistency and avoid the appearance degradation problem in video inapinting task.

- Guided by the consistent context video, video diffusion model can focus on the problem of short-range dynamics modeling, enabling improved motion stability and temporal coherence.

- Benefited from the decomposition of appearance preservation and motion modeling, Vid-Splice can generate vivid motion dynamics with consistent identity.

## 3.1 CoSpliced Module: Multi-frame Consistent Inpainting

Prior work Bian et al. (2025) has demonstrated that inpainting the first frame of a video and subsequently modeling the motion within the masked regions using an I2V generation model yields more effective and higher-quality results than directly performing inpainting across the entire sequence Zi et al. (2025). Inspired by this insight, we extend the first-frame-inpainting paradigm by proposing CoSplice Module to inpaint contextual frame intervals, which diffuses the content of initial reference frame into subsequent sampled frames through a splicing mechanism. Specifically, the CoSplice Module consists of three stages: sampled video formation, reference-guided inpainting and spliced video formation.

**Sampled Video Formation.** While a full video sequence contains rich contextual information, it also contains unnecessary redundancy for video inpainting task. To address this, we apply a temporal sampling strategy with a stride of $K$, selecting $\lceil T/K \rceil$ key frames from the original sequence. These key frames capture salient inter-frame variations in viewpoint and scene dynamics, thereby providing

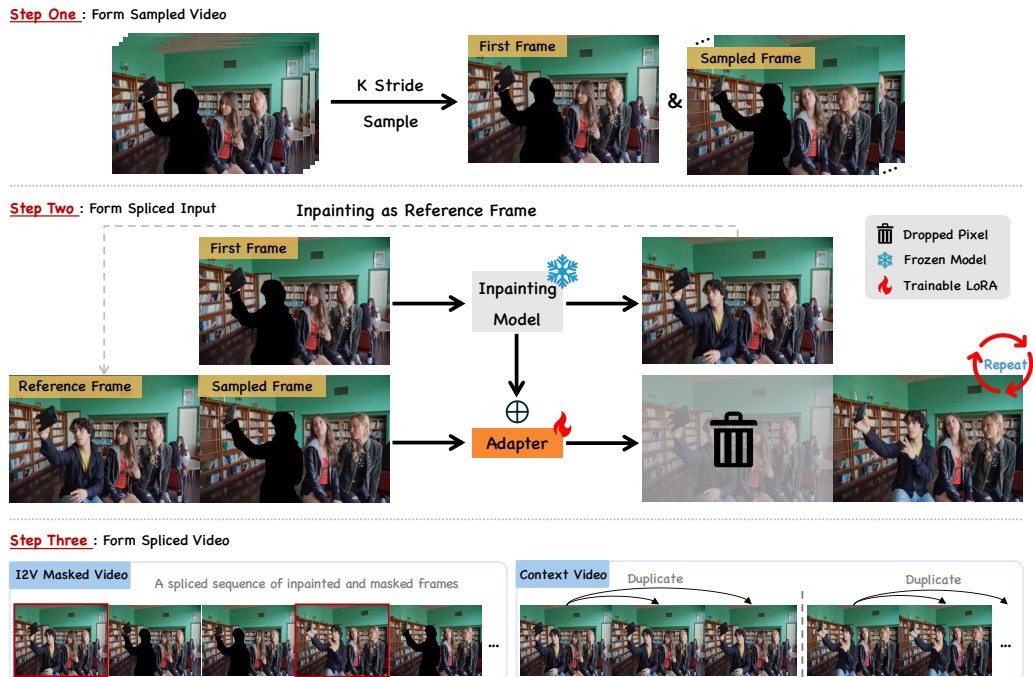

Figure 3: The pipeline of the CoSpliced module. It generates both the I2V-masked video and the context video through three steps: Sampled Video Formation, Reference-guided Inpainting, and Spliced Video Formation.

informative guidance for subsequent video inpainting while remaining compactness. Meanwhile, to mitigate temporal inconsistencies caused by occlusion, we precede the inpainting stage with an optical flow completion Zhou et al. (2023) which only takes few seconds. This allows known pixels to be warped to fill the mask, effectively shrinking the mask area.

**Reference-guided Inpainting.** After temporal sampling, we perform inpainting on both the first frame and the sampled frames. Due to viewpoint variations across frames, inpainting all frames independently leads to discrepancies in structure and appearance within frames. As illustrated in **Step Two** of Fig. 3, we introduce a novel splicing mechanism to enforce identity alignment across frames. Specifically, we train a low-rank ID adapter $\phi$ on image inpainting model $\Psi$ for ID alignment:

$$h = Wx + B_\phi A_\phi x, \quad B_\phi \in \mathbb{R}^{d \times r}, A_\phi \in \mathbb{R}^{r \times d}, \tag{1}$$

where $x$ and $h$ indicate input and output features with channel $d$ for the linear layers in attention blocks; $W$ denotes the frozen weights of original inpainting model, while $B_\phi, A_\phi$ are trainable low-rank matrices with much fewer parameters compared to $W$, *i.e.*, $r \ll d$. Once the first frame is inpainted, it is concatenated along the width dimension inspired by Cao et al. (2024) with each sampled frame and fed into the image inpainting model with adapter. This spliced pixel-level generation mechanism is highly effective for appearance preservation, as shown in Fig. 1, where the reference context flows naturally into the target masked region. The ID adapter is trained on a multi-view dataset and can be broadly applicable to various downstream tasks. The training process can be formulated as follows:

$$\mathcal{L}_{ID}(\phi) = \mathbb{E}\left[\|(\varepsilon - x_0) - \Psi_\phi([x_t; \widetilde{x}_0; M], P, t)\|^2\right] \tag{2}$$

where $\varepsilon$ is noise, $x_0$ is the groundtruth, $\widetilde{x}_0$ is masked frame, $x_t$ is noised frame, and $M$ is image mask. Furthermore, benefited from the strong associative capabilities of state-of-the-art image inpainting model, our method effectively injects novel view synthesis ability into the VidSplice pipeline. As illustrated in the red car example in Fig. 1, VidSplice is able to plausibly infer and synthesize unseen regions from new viewpoints, significantly enhancing spatial consistency and realism. Notably, in scenarios involving long-time motion, we inpaint the first frame and then partition the video into

several temporal chunks. For each identity-aligned chunk, the last inpainted frame is used as first frame for guiding the inpainting of the subsequent chunk to keep inter-chunk consistency.

**Spliced Video Formation.** The contextual frame intervals are utilized as anchors to construct the new I2V masked video $V_m$ and context video $V_c$, as illustrated in **Step Three** of Fig. 3. For the I2V-masked video, we directly insert the inpainted frames back into their original temporal positions, while leaving the remaining masked frames unchanged. Besides, we construct I2V video mask $M^{i2v}$ with the original input mask, expect the mask of first frame which is set to all zero. The resulting sequence and mask serve as the input to the I2V branch. In parallel, we construct the context video via temporal duplication, repeating each inpainted keyframe to fill the subsequent interval until the next, thus matching the original frame count. The context video is sent to the context controller paired with an all zeros mask $M^0$. It should be notice that a mask value of 1 indicates a region that can be modified, while a value of 0 denotes fixed content.

## 3.2 CONTEXT GUIDANCE VIA CONTROLLER INJECTION

We introduce a context controller to guide the generation of diffusion model with context video. The controller is initialized with pretrained diffusion transformer blocks. Inspired by Bahmani et al. (2024); Liang et al. (2024b), VidSplice context controller injects control information only into the first 10 layers of the I2V diffusion transformer, which captures low-level camera and motion information. For the context video $V_c$ and I2V masked video $V_m$ generated in Sec. 3.1, we first employ a 3D VAE encoder to get their latent features $f_c$ and $f_m$, respectively. The latent feature is then concatenated along the channel dimension with a randomly initialized noise and corresponding video masks. Following ControlNet Zhang & Agrawala (2023), the feature of context video is then passed through a zero-initialized learnable embedding layer $\mathcal{F}$. The context controller $\mathcal{H}$ accepts this zero-initialed embedded feature and embedding of I2V video as input and get the control feature $f$ as follows:

$$f = \mathcal{H}\left(\mathcal{F}\left(\left[f_c; \epsilon; M^0\right]\right) + \mathcal{F}_m\left(\left[f_m; \epsilon; M^{i2v}\right]\right)\right) \tag{3}$$

where $[\cdot]$ denotes concatenation operation, $\mathcal{F}_m$ is the pretrained embedding layer of I2V branch. Our formulation ensures that the context latent captures motion variations across frame intervals while remaining stable throughout the forward diffusion process.

## 3.3 MASKED AREA MOTION PROPAGATION

For the I2V masked video generated in Sec. 3.1, we similarly concatenate it with randomly initialized noise and its corresponding mask along the channel dimension and send it to diffusion transformer (DiT). After obtaining the context feature $f_c$ from Eq. 3, we add it into the output feature of corresponding layers of DiT during the forward pass. We employ flow matching loss Lipman et al. (2022) for the optimization of context controller while freeze the diffusion transformer:

$$\mathcal{L}_{video}\left(\mathcal{H}, \mathcal{F}\right) = \mathbb{E}\left[\left\|\left(\varepsilon - \hat{f}^0\right) - \text{DiT}\left(\left[\hat{f}^t; f_m; M^{i2v}\right], f, P, t\right)\right\|^2\right], \tag{4}$$

where $\hat{f}^0$ is the latent feature of groundtruth video $V$, $\hat{f}^t$ represents the noised video latent of $\hat{f}^0$.

## 4 EXPERIMENTS

**Setups**. VidSplice is built upon the Wan2.1 Wang et al. (2025) I2V-14B. The training process consists of two stages: In the first stage, we train an ID alignment adapter based on FLUX.Fill Labs (2024) with a LoRA rank of 128 at a resolution of $512 \times 512$. We use the AdamW optimizer with a learning rate of 1e-4, a batch size of 16, and train for 80,000 steps. The sampling step is set to 50 to enhance performance across different inpainting tasks. In the second stage, we construct the CoSpliced module using the pretrained adapter and proceed to train the transformer and the context controller jointly. This stage is conducted at a resolution of 480P, with a batch size of 1, using AdamW with a learning rate of 1e-5. We utilize 4 H100 GPUs to run all experiments.

**Dataset.** To enable effective ID alignment in the CoSpliced module, we utilize a multi-view dataset. Specifically, we use image pairs from MegaDepthLi & Snavely (2018), which contains diverse multi-view scenes of landmarks. For training, masks are constructed from a combination of randomly

Figure 4: The inpainting comparison between VidSplice, ProPainter, DiffuEraser and VideoPainter.

| | | Video Inpainting | | | | | | Video Editing | | | | |
|---|---|---|---|---|---|---|---|---|---|---|---|---|
| | Models | PSNR↑ | SSIM↑ | LPIPS↓ | MSE↓ | CLIP Sim↑ | Models | PSNR↑ | SSIM↑ | LPIPS↓ | MSE↓ | CLIP Sim↑ |
| VPBench-S | ProPainter | 20.97 | 0.87 | 9.89 | 1.24 | 17.18 | UniEdit | 9.96 | 0.36 | 56.68 | 11.08 | 14.23 |
| | COCOCO | 19.27 | 0.67 | 14.80 | 1.62 | 20.03 | DitCtrl | 9.30 | 0.33 | 57.42 | 12.73 | 15.59 |
| | Cog-Inp | 22.15 | 0.82 | 9.56 | 0.88 | 21.27 | ReVideo | 15.52 | 0.49 | 27.68 | 3.49 | 20.01 |
| | VACE | 25.77 | 0.81 | 7.94 | 0.47 | 18.86 | VACE | 26.14 | 0.80 | 7.63 | 0.49 | 17.02 |
| | VideoPainter | 23.32 | 0.89 | 6.85 | 0.82 | 21.49 | VideoPainter | 22.63 | 0.91 | 7.65 | 1.02 | 20.20 |
| | VidSplice | 28.22 | 0.85 | 5.95 | 0.24 | 21.24 | VidSplice | 28.73 | 0.86 | 6.86 | 0.20 | 20.64 |
| VPBench-L | ProPainter | 20.11 | 0.84 | 11.18 | 1.17 | 17.68 | UniEdit | 10.37 | 0.30 | 54.61 | 10.25 | 15.42 |
| | COCOCO | 19.51 | 0.66 | 16.17 | 1.29 | 20.42 | DitCtrl | 9.76 | 0.28 | 62.49 | 11.50 | 16.52 |
| | Cog-Inp | 19.78 | 0.73 | 12.53 | 1.33 | 21.22 | ReVideo | 15.50 | 0.46 | 28.57 | 3.92 | 20.50 |
| | VACE | 24.57 | 0.75 | 10.35 | 1.00 | 20.45 | VACE | 21.70 | 0.70 | 12.36 | 1.12 | 17.03 |
| | VideoPainter | 22.19 | 0.85 | 9.14 | 0.71 | 21.54 | VideoPainter | 22.60 | 0.90 | 7.53 | 0.86 | 19.38 |
| | VidSplice | 28.11 | 0.84 | 5.31 | 0.26 | 22.32 | VidSplice | 26.17 | 0.82 | 8.11 | 0.28 | 19.55 |

Table 1: Inpainting and editing evaluation on VPBench. Deeper color indicate better performance. Note we scale LPIPS and MSE metrics 100× for convenient comparison.

generated masks and semantic masks from the COCO dataset with a masking ratio ranging from 40% to 70%. In total, we curate a dataset of 820K masked image pairs for training the adapter. For training the embedding layer and context controller, we select the VPData dataset introduced by VideoPainter Bian et al. (2025), which consists of approximately 450K publicly available internet videos covering a wide range of indoor and outdoor scenes. For evaluation, we conduct experiments on both the widely used DAVIS dataset Perazzi et al. (2016) and the VPBench benchmark (VPBench-S and VPBench-L) proposed by VideoPainter.

**Metrics.** We evaluate the perseverance of the unmasked regions in the generated video using metrics including PSNR, SSIM, LPIPS, and MSE. For a clearer comparison of the results, we scale both LPIPS and MSE metrics 100×. To assess the semantic alignment between the generated content within the masked regions and the textual prompt, we employ the CLIP similarity Radford et al. (2021) score as a quantitative measure. We also evaluate FVID metric for overall video quality.

## 4.1 RESULTS OF VIDEO INPAINTING

**Quantitative results.** As shown in Tab. 1 and Tab. 2, we conduct quantitative comparisons with state-of-the-art video inpainting methods. Both VACE Yang et al. (2024) and VideoPainter demonstrate strong generative capabilities within the masked regions, as reflected by their high CLIP similarity scores. Our proposed VidSplice also achieves competitive performance in terms of content generation, particularly on the VPBench-L benchmark. This can be attributed to the incorporation of anchor frames and context video, which effectively constrain the generation process. In addition, VidSplice attains competitive results on background preservation metrics, demonstrating its ability to maintain the integrity of unmasked content while performing high-quality foreground inpainting.

**Qualitative results.** We visualize the qualitative results of state-of-the-art methods—ProPainter Zhou et al. (2023), DiffuEraser Li et al. (2025), VideoPainter Bian et al. (2025), and our proposed Vid-Splice—on representative video inpainting tasks, as shown in Fig. 4. Benefiting from the incorporation

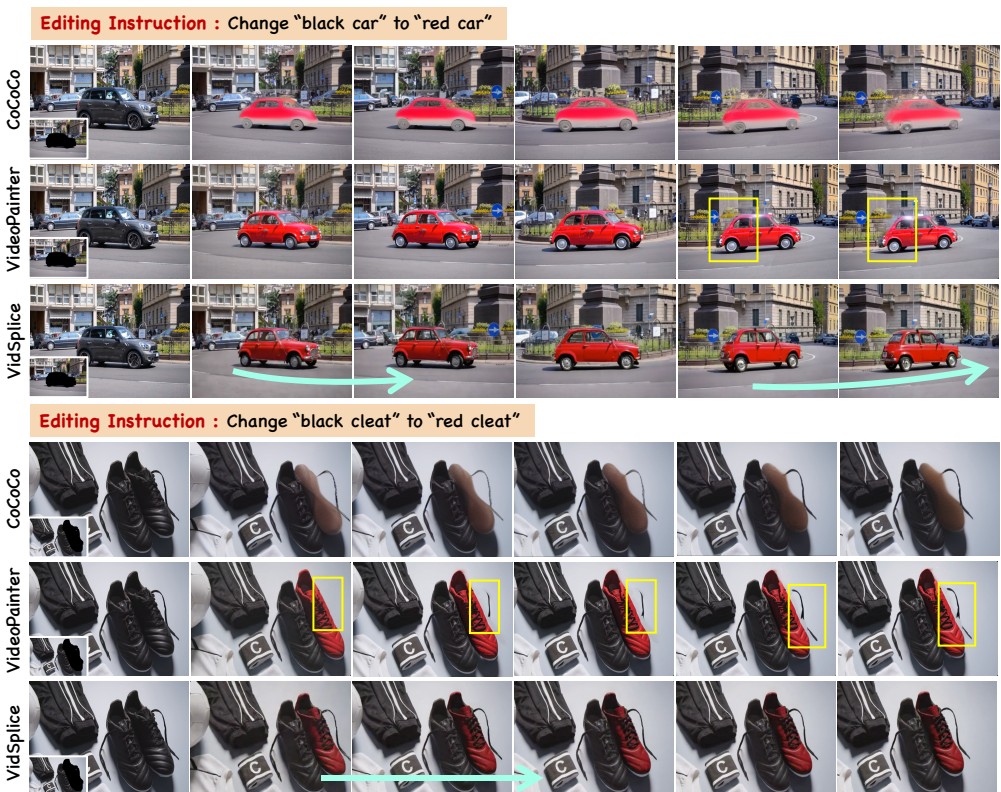

Figure 5: The frame-by-frame editing comparison between VidSplice, ProPainter and COCOCO.

of both inpainting priors and context priors, VidSplice exhibits stable object removal and successfully reconstructs occluded regions such as grass and rail tracks within the masked areas. ProPainter, which is based on a transformer architecture, effectively removes the target region but struggles to generate semantically consistent content in occluded areas. DiffuEraser, initialized Gaussian noise with ProPainter's output, introduces noticeable blurriness in the inpainted regions due to imperfect mask-aware generation. VideoPainter, on the other hand, suffers from severe artifacts in removal scenarios: residual motion are still visible inside the masked area, and only partial alignment with the background is achieved, as highlighted in the red box in Fig. 4. Additionally, VideoPainter exhibits unfilled regions (see blue boxes), likely due to insufficient learning of its context branch. In contrast, VidSplice maintains both spatial coherence and temporal consistency, demonstrating more faithful reconstruction and stable content propagation across frames.

## 4.2 RESULTS OF VIDEO EDITING

**Quantitative results.** Following VideoPainter, we conduct quantitative comparisons against recent video editing models including UniEdit Bai et al. (2024), DitCtrl Cai et al. (2024), and ReVideo Mou et al. (2024), as summarized in Tab. 1. Thanks to the powerful I2V backbone, VidSplice consistently outperforms competitors on background preservation metrics across both short videos (VPBench-S) and long videos (VPBench-L), with particularly strong performance in PSNR. VideoPainter benefits from training on higher-resolution videos, which gives it great SSIM metrics. Notably, VidSplice still achieves top-tier CLIP similarity scores, highlighting the effectiveness of injected contextual frame intervals in enhancing semantic alignment with the prompt. In summary, VidSplice achieves state-of-the-art quantitative performance in both semantic alignment of foreground generation and background consistency preservation.

**Qualitative results.** We compare VidSplice with representative video editing methods that accept masked video inputs, namely COCOCO and VideoPainter, on both VPBench and DAVIS, as visualized in Fig. 5. The selected examples represent two common scenarios: one involving object motion, and the other involving camera motion. Similar to VideoPainter, VidSplice leverages the inpainted

| | Models | PSNR↑ | SSIM↑ | LPIPS↓ | MSE↓ | CLIP Sim↑ |
|---|---|---|---|---|---|---|
| DAVIS | ProPainter | 26.05 | 0.73 | 9.90 | 1.21 | 15.02 |
| | DiffuEraser | 25.21 | 0.73 | 9.72 | 1.22 | 15.25 |
| | COCOCO | 21.34 | 0.66 | 10.51 | 0.92 | 17.50 |
| | Cog-Inp | 23.92 | 0.79 | 10.78 | 0.47 | 17.53 |
| | VideoPainter | 25.27 | 0.94 | 4.29 | 0.45 | 18.46 |
| | VidSplice | 27.34 | 0.82 | 9.15 | 0.25 | 20.54 |

Table 2: Inpainting evaluation on DAVIS.

| Models | PSNR↑ | SSIM↑ | LPIPS↓ | MSE↓ | CLIP Sim↑ |
|---|---|---|---|---|---|
| Stride=5 | 26.57 | 0.81 | 8.57 | 0.30 | 20.48 |
| Stride=20 | 26.52 | 0.81 | 8.99 | 0.30 | 20.49 |
| Layers=2 | 22.05 | 0.75 | 11.01 | 0.81 | 20.34 |
| Layers=5 | 26.34 | 0.81 | 8.66 | 0.32 | 20.54 |
| w/o ID Align | 25.23 | 0.78 | 10.95 | 0.59 | 20.87 |
| VidSplice | 27.34 | 0.82 | 9.15 | 0.25 | 20.54 |

Table 3: Ablation study evaluation on DAVIS.

| Methods | ProPainter | COCOCO | UniEdit | DitCtrl | ReVideo | VACE | VideoPainter | VidSplice |
|---|---|---|---|---|---|---|---|---|
| FVID↓ | 0.44 | 0.69 | 1.36 | 0.57 | 0.42 | 0.20 | 0.18 | 0.16 |

Table 4: Comparison of FVID metrics for video quality on VPBench.

first frame as a strong anchor, achieving high-quality edits at the beginning of the video. In contrast, although COCOCO produces semantically correct edits, its results often suffer from inferior visual quality and temporal inconsistency across the entire video. In the "red car" case from Fig. 5, VideoPainter exhibits content degradation toward the end of the video. In contrast, VidSplice maintains high fidelity, thanks to its multi-frame consistent inpainting priors introduced by the CoSpliced module, which enable more robust generation even under large object motions and novel view transitions. In the "red cleat" case, VidSplice demonstrates stable inpainting performance, while VideoPainter exhibits temporal flickering, particularly around the "shoelace" region. We also summarize the Fréchet Video Distance (FVID) scores for overall video quality and motion smoothness in Tab. 4, which demonstrate that VidSplice, VideoPainter and VACE achieves competitive generative performance within the masked region.

## 4.3 ABLATION ANALYSIS

In our proposed VidSplice framework, the sample stride serves as a key hyperparameter that determines how many contextual frame intervals are pre-inpainted and injected into the video generation process. Another crucial parameter is the number of controller layers, which directly influences the model's capacity to capture camera and motion information. Additionally, we conduct ablation on the CoSplice Module to evaluate the effectiveness of the ID alignment strategy introduced for enhancing frame consistency. All results are summarized in Tab. 3.

**Sample Stride.** As shown in row 1, 2, and 6, selecting a stride of 10 achieves not only high efficiency but also superior performance. This demonstrates that too few or too many image priors can degrade visual quality and prompt following. Our VidSplice effectively strikes a balance.

**Controller Layers.** As shown in the 3rd, 4th, and 6th rows of Tab. 3, reducing the number of context controller layers leads to a sharp drop in performance, demonstrating the effectiveness of our controller design. This can be attributed to fewer injection layers limit the I2V model's ability to learn context and motion information.

**CoSpliced Modulde.** When the ID alignment adapter is removed and unpainted frames are directly fed input into the image inpainting model, we observe an increase in CLIP similarity in Tab. 3 row 5. However, this comes at the cost of significantly reduced temporal consistency in the output video and detailed qualitative results of ID alignment can be found in the supplementary material. This provides strong evidence for the effectiveness of our CoSplice module, which ensures consistent foreground inpainting during video generation.

## 5 CONCLUSION

This paper presents VidSplice, a new video inpainting framework that separates spatial and temporal aspects into two tasks: multi-frame inpainting and motion propagation. It introduces a CoSpliced module and a lightweight context controller to better use frame and motion cues for improved video generation. The CoSpliced module aligns content across frames, maintaining foreground quality even with large movements. Unlike previous methods that depend on single-frame input and implicit motion learning, VidSplice allows for better content control. Together, these components enhance inpainting quality and reduce distortion across diverse video scenes.

## ETHICS STATEMENT

This paper explores video inpainting using video and image generation models. While these models are powerful, they can also produce misleading or fake content. Users should be mindful of this. Privacy and consent are also important, as these models are trained on large datasets, which may include biases that can lead to unfair results. We encourage responsible and inclusive use of such models. Note that our work focuses purely on technical aspects, and all pre-trained models used are publicly available.

## REPRODUCIBILITY STATEMENT

We are committed to ensuring the reproducibility of all results reported in this paper, in accordance with ICLR standards. To this end, we will publicly release our codebase, pre-trained model checkpoints, and all scripts necessary to reproduce our experiments and main results upon publication.Comprehensive details regarding model architecture, training procedures, hyperparameters, and dataset preprocessing are provided in main paper to facilitate independent verification.

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

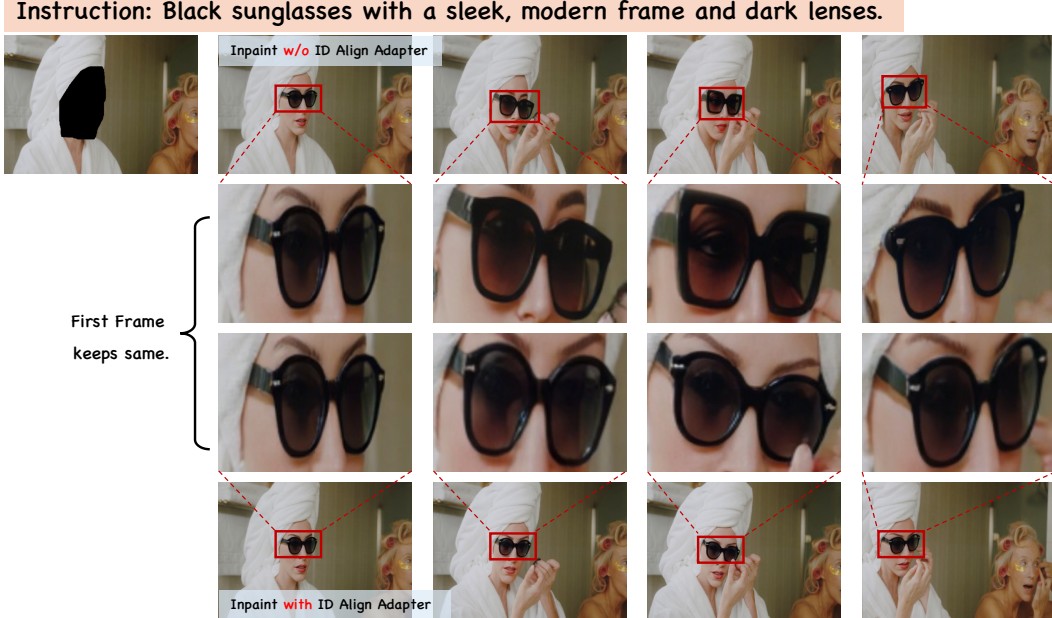

Figure 6: Qualitative ablation results of ID Alignment Adapter in VidSplice framework.

# APPENDIX FOR VIDSPLICE

For a better understanding of our work, we have provided additional analysis and visualizations, which are organized as follows:

- **A    Ablation Study of ID Alignment Adapter**
  This section qualitatively showcases the effectiveness of the ID Alignment Adapter in the CoSplice Module for identity alignment/preservation.

- **B    Discussion about Inference Time**
  This section discusses the comparison of inference time between our method and our baseline, VideoPainter, which is also an Image-to-Video (I2V) model.

- **C    More qualitative results of VidSplice**
  In this section, we provide additional qualitative results to support VidSplice's strong performance, including video object removal and editing.

- **D    Discussion about Challenging Scenarios**
  This section addresses more challenging and practical scenarios, such as: the Fully Occlusion Scenario, the Fast-Moving Object and Dynamic Shot Scenario, and Perspective and Object Dynamics.

- **E    User Study**
  We provide a simple user study to support the quantitative and qualitative metrics in the main paper.

- **F    Usage of Large Language Models**
  A statement regarding our utilization of Large Language Models in this research.

## A    ABLATION STUDY OF ID ALIGNMENT ADAPTER

As shown in Fig. 6, we further provide qualitative results to assess the effectiveness of the ID Alignment Adapter. The first and fourth rows correspond to inpainting results *without* and *with* the adapter, respectively, while the second and third rows present zoomed-in views of key regions. It is evident that, under identical random seeds, the absence of the adapter leads to noticeable identity

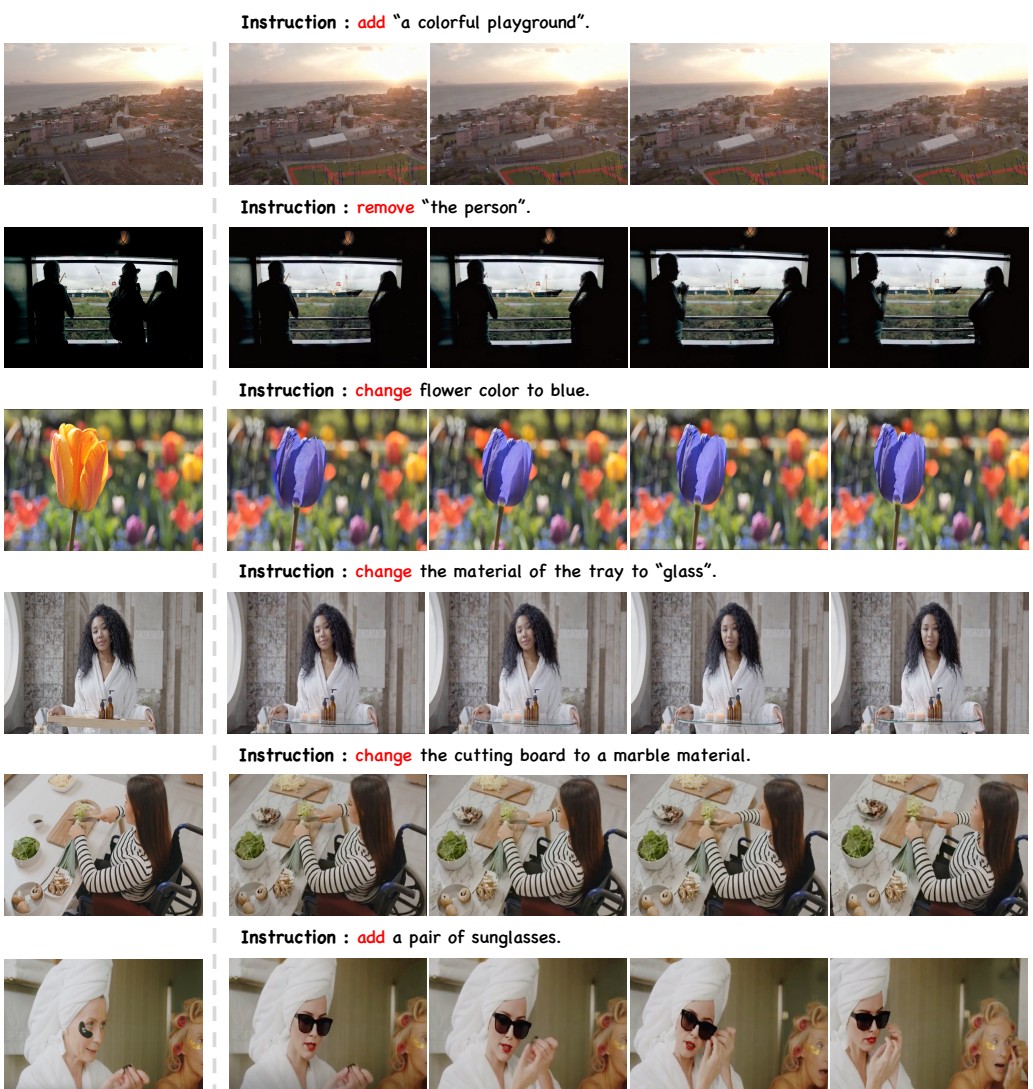

Figure 7: More qualitative results of VidSplice in video inpainting and video editing tasks.

distortion—e.g., a circular sunglass is inpainted as a square one. In contrast, the model equipped with the ID Alignment Adapter consistently preserves instance-level appearance, demonstrating its crucial role in maintaining temporal coherence and identity fidelity in VidSplice.

## B    DISCUSSION ABOUT INFERENCE TIME

Regarding efficiency, we conducted a comparative analysis of inference time with VideoPainter, which is also capable of handling both video inpainting and editing tasks. Both VidSplice and VideoPainter are built upon the Image-to-Video (I2V) concept. The pipeline's runtime includes the generation of the entire video, incorporating the initial frame's generation time. Notably, our VidSplice pipeline also encompasses the generation of anchor frames through reference inpainting.

When processing a 49-frame video, the VideoPainter pipeline has an inference cost of approximately 2.55 seconds per frame. In contrast, our VidSplice can process a greater number of frames, up to 81, in a single inference, with a total inference cost of about 2.59 seconds per frame, which is comparable to VideoPainter. It is important to note that VideoPainter is constrained by the limitations of its base

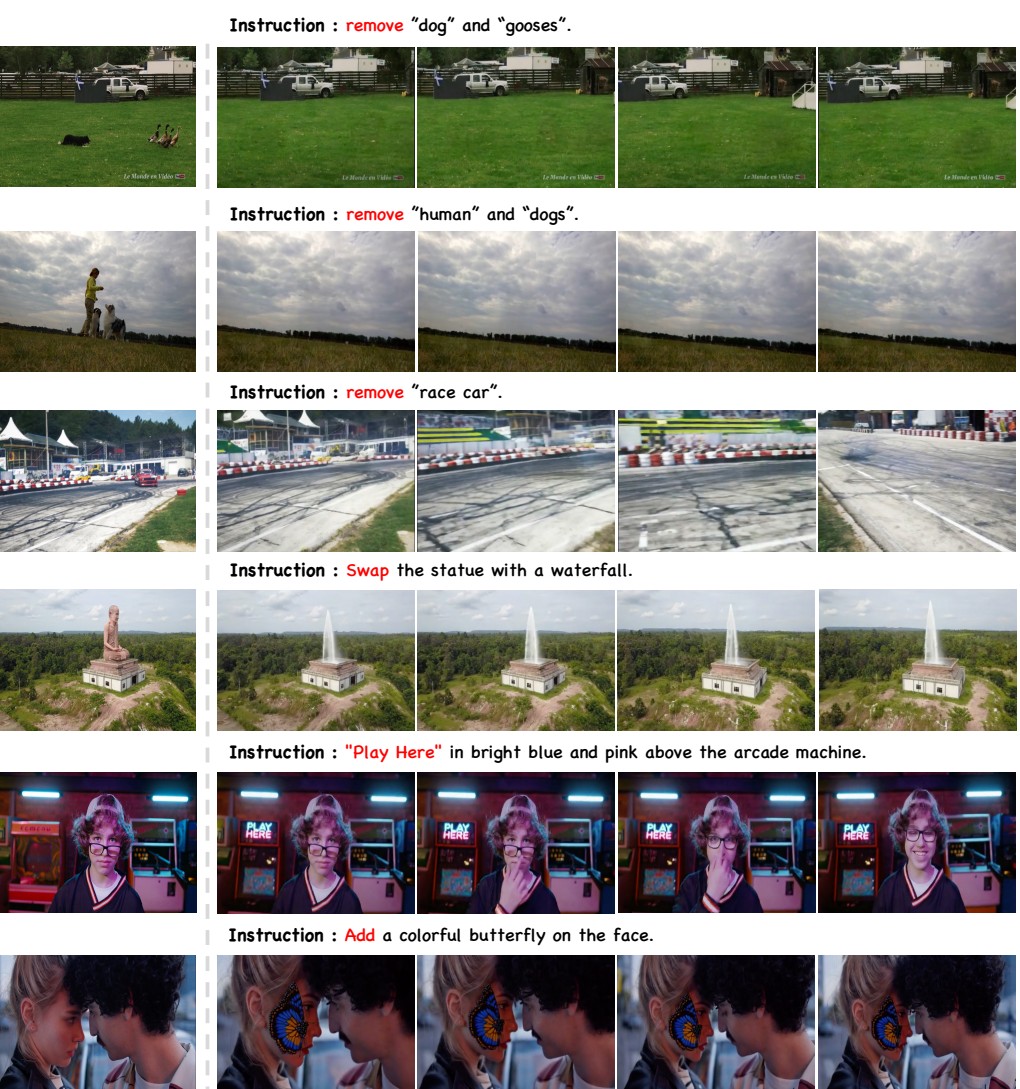

Figure 8: More qualitative results of VidSplice in video inpainting and video editing tasks.

model, with a processing cap of 49 frames per video. This necessitates more frequent segmentation of video clips and multiple rounds of inference when processing real-world video data. This, in turn, would significantly increase the total processing time. Furthermore, because VideoPainter lacks mechanisms such as reference inpainting to maintain content consistency between clips, it introduces the latent issue of ID inconsistencies between preceding and succeeding clips. All comparative experiments were conducted using the same number and model of GPUs, and the model used for generating the initial frame was FLUX.1.Fill in all cases.

## C    MORE QUALITATIVE RESULTS OF VIDSPLICE

Fig. 7 and Fig. 8 presents additional qualitative results of VidSplice. The first column shows the first frame of the original video, and the remaining columns depict generated frames. The top four rows are sampled from the DAVIS dataset, which focuses on inpainting and object removal. As shown, VidSplice achieves effective removal across various scenes. The bottom rows are from VPBench, covering diverse editing scenarios: changes in object attributes (Rows 5 and 8), text generation (Row 6), and face identity preservation after modification (Rows 7 and 9). These results

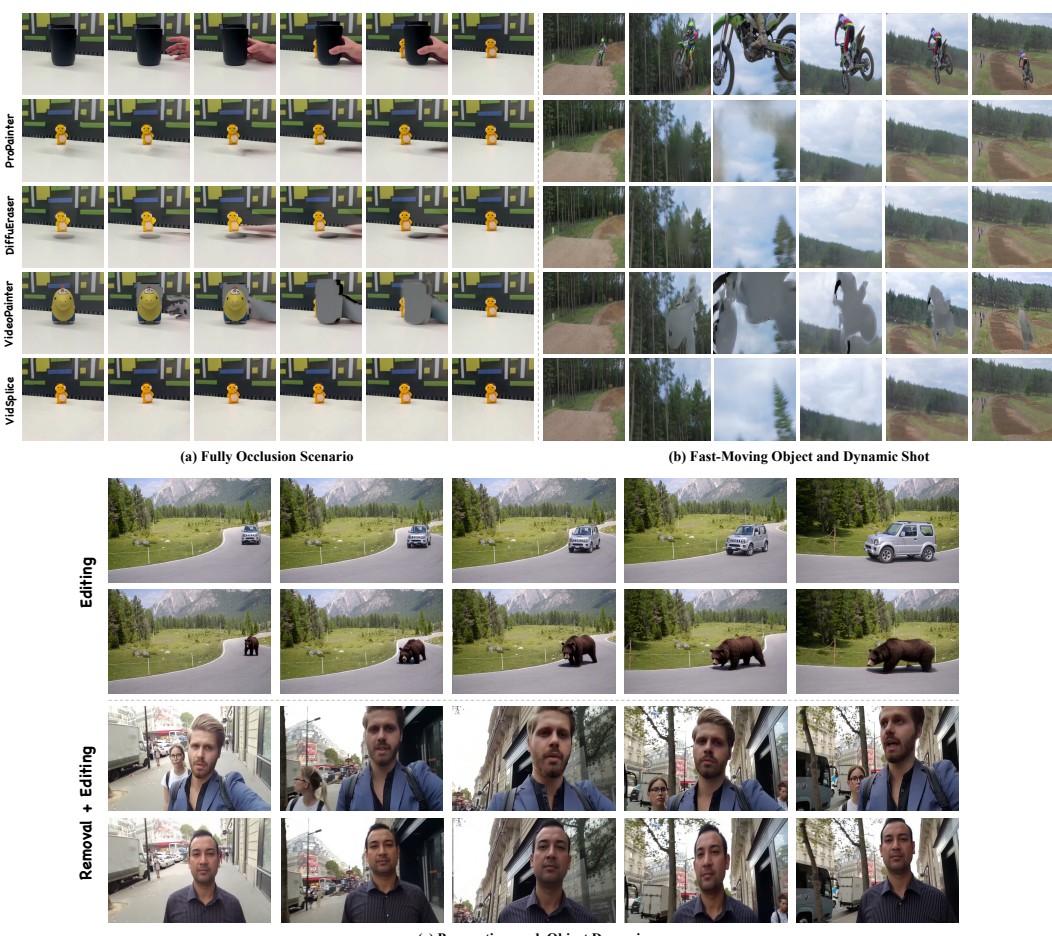

(a) Fully Occlusion Scenario        (b) Fast-Moving Object and Dynamic Shot

(c) Perspectives and Object Dynamics

Figure 9: More challenging and practical qualitative visualization of (a) fully occlusion scenario, (b) fast-moving object with dynamic shot scenario and (c) perspectives and object dynamics.

demonstrate VidSplice's strong performance in both video inpainting and video editing tasks. We have already provided a qualitative comparison between VidSplice and VideoPainter on the object removal task in the main text. Here, we present a comparison between VidSplice and VACE in Fig 10. VidSplice consistently outperforms VACE MV2V version without introducing additional artifacts, while maintaining spatial coherence and temporal consistency.

## D  DISCUSSION ABOUT CHALLENGING SCENARIOS

**Fully Occlusion Scenario.** Image-to-Video (I2V) models, such as VideoPainter, are capable of addressing a wide range of video editing tasks by utilizing an edited image and a corresponding mask. However, these models often falter in object removal tasks, especially when encountering scenarios with full occlusion. This frequently results in inference failure due to a lack of frame-to-frame consistency. To overcome this challenge, our proposed method, VidSplice, integrates the optical flow completion results from ProPainter during the inference stage to guide pixel warping. This differs from DiffuEraser, which uses the complete output from ProPainter to initialize the diffusion noise. Thanks to ProPainter's lightweight architecture, VidSplice can acquire partial pixel information for occluded regions via optical flow within seconds. Concurrently, our use of anchor frames with reference inpainting ensures the effective propagation of identity information, thereby maintaining inter-frame consistency. We present a visual comparison in Fig. 9 (a) that demonstrates the performance of different optical flow-based methods in a fully occluded scenario.

**Instruction:** Stone courtyard with steps, flower arrangements on the wall, and paved ground.

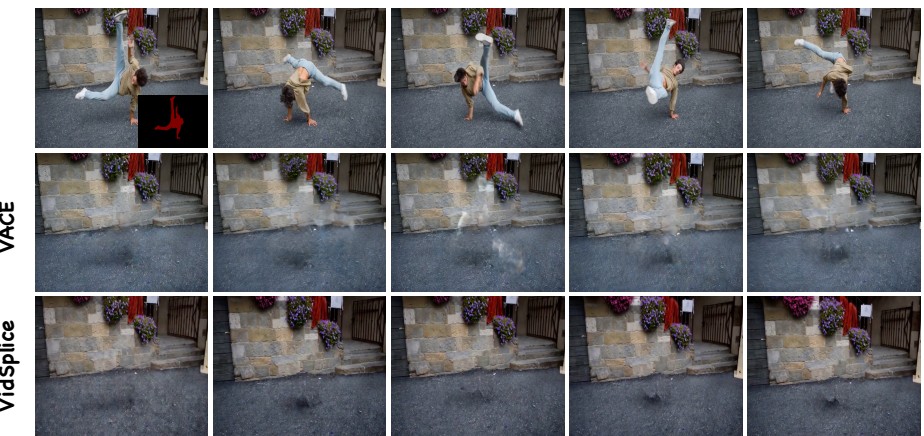

**Instruction :** A colorful graffiti mural on a wall with the word "LIMITS" in blue and pink shades, two small windows above, red brick walls on the sides, grassy foreground, and a doorway on the right.

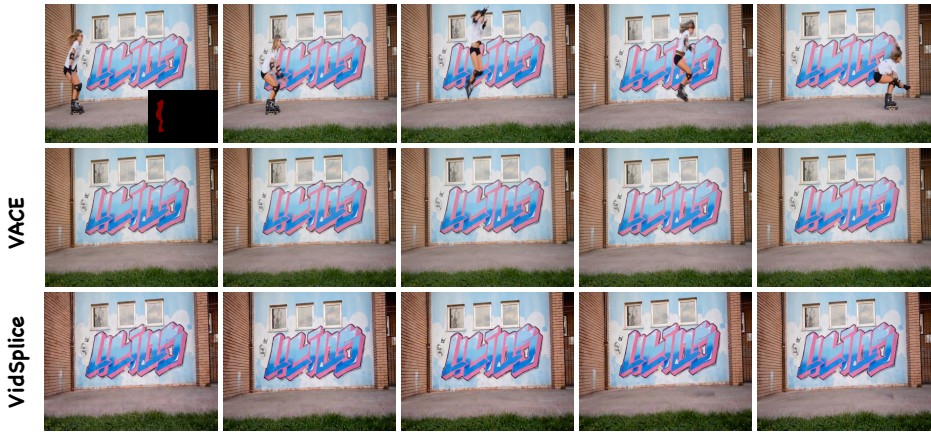

**Instruction :** A terrarium with a branch, climbing ropes, rocks, and leafy plants.

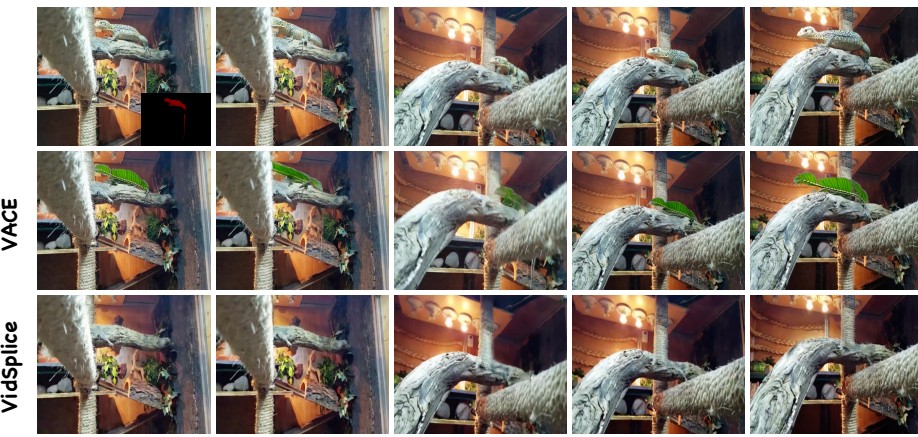

Figure 10: More qualitative comparison between VidSplice and VACE.

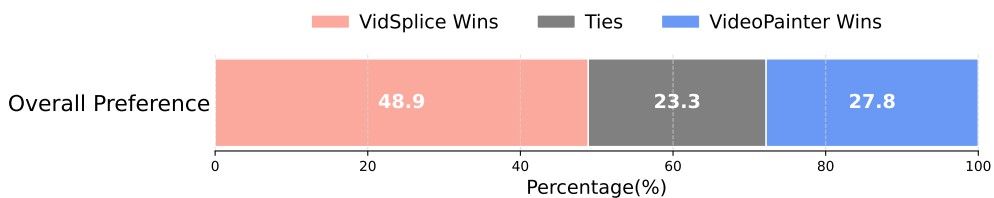

Figure 11: User study of VidSplice and VideoPainter on DAVIS benchmark.

**Fast-Moving Object and Dynamic Shot Scenario.** Another challenging scenario in practical applications is the presence of fast-moving objects and dynamic shots, which demands superior temporal stability and scene consistency from the model. We use the "motocross-jump" sequence from the DAVIS dataset as an example to evaluate performance under these demanding conditions. This scene features a motorcycle executing a high-speed jump, coupled with significant camera motion. As illustrated in Figure 9 (b), we also provide a comparative analysis of the inference results from ProPainter, DiffuEraser, VideoPainter and our VidSplice. Both ProPainter and DiffuEraser can achieve great or acceptable object removal results, while VideoPainter often exhibits inferior inpainting performance when confronted with such dynamic scenarios. This limitation in VideoPainter can be attributed to the lack of contextual information injection faces with complex physical and motion modeling. Our VidSplice achieves competitive object removal results without inference time optical flow warping, even in this highly dynamic scenario.

**Perspective and Object Dynamics.** Addressing different perspective object dynamics is an crucial problem for video editing. The ID alignment adapter within our CoSplice module is specifically designed for this, benefiting from its training on multi-view data. As we have shown some cases in the main text, the adapter can generate objects from new viewpoints without requiring explicit conditions like camera poses. To further substantiate this capability, here we present additional intermediate results from CoSplice that showcase its effectiveness in generating plausible perspectives in Figure 9 (c). When introducing camera pose for explicit pixel warping, reference inpainting becomes easier because the inpainting/editing region is partial fixed, but we do not consider these additional information in this work.

# E    USER STUDY

To provide a more comprehensive evaluation of the generated video quality, we conducted a user study based on the DAVIS benchmark (includes 90 videos). We invited 20 participants for simple human evaluation. The compared methods include VideoPainter and our VidSplice, which were evaluated on overall preference, spatial coherence, and temporal consistency. The human evaluation results, presented in Fig. 11, indicate that VidSplice is rated superior to VideoPainter on the DAVIS dataset, particularly for the object removal task.

# F    USAGE OF LARGE LANGUAGE MODELS

This article only utilizes a large language model to polish the language descriptions. The prompt used is as follows:

"You are a professional computer science researcher. Please revise the following sentences in the style of ICLR to make them logically coherent."

