# OpenReview forum: "VidSplice: Towards Coherent Video Inpainting via Explicit Spaced Frame Guidance"
_ICLR.cc/2026/Conference — ICLR 2026 Conference Withdrawn Submission_

### Official Review · Reviewer_jSYU · 2025-10-29

**Soundness:** 3
**Presentation:** 2
**Contribution:** 2
**Rating:** 2
**Confidence:** 4

**Summary:**

This paper tackles the problem of motion inconsistency in existing video editing models. The paper proposed to decouple the inconsistency problems into two sub-problems, per-frame spatial coherence and cross-frame temporal consistency with their CoSpliced module and the context controller.  The authors evaluate their proposed adaptive modules with Wan 2.1 and achieve state-of-the-art results on VPBench.

**Strengths:**

- This paper studies an important problem of video inpainting for dynamic scenes.
- The proposed method resulted in better inpainting performance quantitatively and qualitatively in the VPBench.

**Weaknesses:**

**Writing**
- Clarity of writing need to be improved, for example, the overview of Fig.2 in the paper is not helping the readers to understand the contribution.  I.e. what is I2V-masked video, and what is the context video ?

**Experiments**
- Sentences like “Thanks to the powerful I2V backbone, VidSplice consistently outperforms competitors” will totally destroy the credibility of your proposed modules, like the CoSpliced module and the context controller. Make it sound like you are not doing an apple-to-apple comparison with the baseline.

- There is zero mention of the computational comparison between your method and baselines VideoPainer and VACE in the main paper. How big is the total number of parameters of your adaptive modules and theirs ? Training cost/inference cost, etc

- The ID alignment module is critical to the success of the proposed approach, yet there is no quantitative comparison of with/without the module. How do the metrics look in Table 1 without the ID alignment module?

- All of the examples shown in the paper have a single mask on a single object, no demonstration of multiple in-painting objects, which raises the concern of the scalability of the proposed approach.

**Questions:**

Please see weakness.

Does the proposed approach work when you have multiple moving objects in the scene ?

The VPBench says the benchmark consists of 100 short videos and 16 long videos, did you evaluate your model/baselines with this amount of data points?

---

### Official Review · Reviewer_YfAK · 2025-10-31

**Soundness:** 2
**Presentation:** 1
**Contribution:** 2
**Rating:** 2
**Confidence:** 3

**Summary:**

This paper proposes VidSplice, a pipeline for video inpainting. The idea is to include additional modules to ensure long-term consistency, while maintaining short-term quality smoothness. The VidSplice pipeline is built around a pre-trained Image-to-Video (I2V) diffusion transformer. To guide this backbone model, the authors introduce several components. These modules inclode the CoSpliced module, the Context Controller, and an optical-flow based model. This modules are themselves composed of more modules.

The method is evaluated on video inpainting and editing tasks using benchmarks like DAVIS and VPBench, where it demonstrates competitive performance against baselines considered.

**Strengths:**

The paper tackles the critical challenge of long-range temporal consistency in video inpainting. The core idea of decoupling the problem and using a set of consistently inpainted anchor frames as explicit guidance is a sensible approach. This "spaced-frame guidance" is a clear departure from methods that rely on a single reference frame, offering a more structured way to control content over time.

The experimental results are strong. The method shows competitive quantitative performance across multiple datasets (VPBench-S, VPBench-L, DAVIS) and metrics (PSNR, SSIM, LPIPS, FVID). The qualitative results presented in Figure 1, 4, and 5 are compelling, demonstrating robust identity preservation and stable motion generation where other methods fail.

**Weaknesses:**

The paper's primary weakness is its lack of clarity. The VidSplice pipeline is highly complex, integrating multiple pre-trained models and newly proposed modules. The current explanations in the text and figures are confusing, ambiguous, and at times contradictory, making it extremely difficult to understand the method's precise mechanics.

- **Confusing Diagrams**: Figure 2, the main pipeline diagram, is too high-level and does not provide enough detail. Key variables and modules defined in Section 3 (e.g., $f\_c, f\_m, \mathcal{H}, \mathcal{F}$) are not labeled on the figure, making it very difficult to follow the data flow.
- **Module Ambiguity**: It is unclear if the "Adapter" shown in Figure 2 is the same as the "Adapter" detailed within the CoSpliced Module in Figure 3. The one in Figure 3 is explained in detail, but there is no explanation for the one in Figure 2 This creates confusion about the architecture.
- **Unjustified Components**: The paper states that the "Context Video" is formed by temporal duplication, repeating each inpainted keyframe. This adds no new information but significantly increases the computational cost for the Context Controller. The paper provides no justification for this design choice over simply using the unique anchor frames. Further, the context video is paired with an "all zeros mask $M^0$". The purpose of feeding a constant tensor of zeros into the model is not explained and seems redundant.
- **Diffusion Process and noise**: Sections 3.2 and 3.3 and Figure 2 explicitly state that noise is concatenated with the latent features. However, the flow matching loss in Eq. 4 uses $\hat{f}^{t}$, which is defined as the "noised video latent". This notation typically implies a standard diffusion/flow matching noising process, not channel-wise concatenation. This is a critical confusion about how the model is trained.
- A similar confusion exists for the CoSpliced module. Its training loss in Eq. 2 uses $x_t$, a "noised frame" in pixel space. But the main pipeline in Figure 2 only shows noise being added after the 3D VAE encoder, in latent space.
- I think providing an algorihtm clearly stating all the steps, with careful notation would go a long way clarifying the confusions.

While the empirical results of the proposed method look promising, I think the paper, in its current form, is not ready for publication and require a major rewriting effort.

**Questions:**

- **Potential OOD Issues**: The paper does not specify if the 3D VAE Encoder is frozen or fine-tuned. If it is frozen (as is common), would masked video inputs $\tilde{V}$ be an out-of-distribution problem?

- **Context Video Rationale**: What is the technical justification for creating the context video by repeating frames and for concatenating an all-zeros mask $M^{\delta}$?

- **Cyclical Dependency**: Equation 3 implies the Context Controller $\mathcal{H}$ takes an embedding from the I2V branch ($\mathcal{F}_m$) as input, while its output $f$ is simultaneously fed into the I2V branch. This appears to be a cyclical definition, but Figure 2 only shows a one-way arrow. How is this implemented?

- **ID Adapter definition**: Could you provide more intuition for the specific formulation in Eq. 1 ($h=Wx+B_{\phi}A_{\phi}x$)? Intuitively, why should one expect this to provide unique ID information?

- **ID Adapter training**: Lines 262 and 263 state that the "The ID adapter is trained on a multi-view dataset and can be broadly applicable to various downstream tasks". Does this mean the ID adapter is trained on a separate dataset and then placed into the pipeline? If so, why is it shown as a trainable module?

---

### Official Review · Reviewer_TFer · 2025-11-01

**Soundness:** 2
**Presentation:** 3
**Contribution:** 2
**Rating:** 2
**Confidence:** 5

**Summary:**

This paper presents VidSplice, a video inpainting framework that addresses temporal consistency challenges in video generation.
The authors decouple the task into two sub-problems: (1) multi-frame consistent image inpainting via a CoSpliced Module that propagates first-frame content to sampled keyframes through reference-guided inpainting, and (2) masked area motion propagation using a Context Controller that injects spaced-frame priors into an I2V diffusion model. The method builds upon Wan2.1 (14B parameters) and demonstrates competitive quantitative results on VPBench and DAVIS benchmarks.

**Strengths:**

1. Addresses an important and practical problem with clear motivation.
2. Comprehensive experimental evaluation across multiple benchmarks (VPBench, DAVIS) and tasks (inpainting, editing)
3. Clear presentation with well-designed figures and overall paper organization.

**Weaknesses:**

1. Limited Technical Novelty. The paper's main technical components are combinations of existing methods rather than novel inventions, which raises concerns about its novelty for a top-tier venue. For instance, the CoSpliced module's idea of using image inpainting models for anchor frames can be found in FloED (Gu et al. 2024), while the context controller concept closely resembles VideoPainter (Bian et al. 2025). The overall pipeline feels like an assembly of existing techniques rather than a cohesive innovation with clear conceptual breakthroughs.

2. Experiments-Related Concerns
a) The quantitative comparisons are fundamentally flawed due to uncontrolled backbone models. VidSplice uses the substantially stronger Wan2.1-14B with FLUX.Fill for assistance, while baseline methods likely use smaller models (e.g., the paper never specifies VACE's model size). Without controlling for backbone capacity, it is impossible to determine whether performance gains stem from the proposed method or simply from superior base models. Critical ablations are missing: applying VidSplice's approach to baseline backbones, or applying baseline methods to the same Wan2.1 backbone.
b) Inconsistent DAVIS Results and Missing Temporal Metrics
I have concerns about the quantitative experiments, particularly regarding DAVIS. Why do the reported ProPainter numbers differ from those in the original ProPainter paper also be different with VideoPainter paper? Are the mask settings different? The paper provides no explanation. Additionally, critical temporal consistency metric "flow warping error" is absent, which is standard for evaluating temporal coherence in video inpainting. Without these metrics, the paper's core claim of improved temporal consistency is inadequately validated.

3. Optical Flow Related Concerns
a) Self-Contradictory Flow Usage and Missing Ablation.
The paper contains a critical self-contradiction regarding optical flow usage. Line 246 states that ProPainter's flow completion is used for image propagation, yet line 993 (Appendix) explicitly denies that VidSplice uses flow warping. More seriously, ProPainter (Zhou et al. 2023) demonstrates that image propagation via flow warping contributes substantially to temporal consistency, yet VidSplice provides no ablation study to isolate or exclude this prior.
b) Unclear Efficiency Analysis.
While the appendix analyzes efficiency, it remains unclear whether optical flow computation is included in the reported inference time. Moreover, the stated processing time (2.59s per frame) does not meet real-time requirements, limiting practical applicability for interactive editing scenarios.

**Questions:**

Refer to Weaknesses.

---

### Official Review · Reviewer_ZcWv · 2025-11-03

**Soundness:** 2
**Presentation:** 3
**Contribution:** 2
**Rating:** 2
**Confidence:** 4

**Summary:**

This paper focuses on the video inpainting task. It argues that existing methods overly rely on single-frame inpainting results as references, leading to the temporal inconsistencies, such as incomplete regions, content distortion, or a failure to maintain spatiotemporal consistency.  To address this issue, the authors propose a decoupled framework that divides video inpainting into two tasks: maintaining spatial coherence within each frame and ensuring temporal consistency across frames. Specifically, they extend first-frame inpainting into multiple keyframes with certrain strides in CoSplice Module and ensure the consistency between inpainted results with IP-Adapters. Then they inject extracted context features into video diffusion models and train a controller similar with ControlNet for better performance.

**Strengths:**

1. CoSplice Module is demonstrated effective by the experiments.
2. This paper provides a very comprehensive comparison with related work。
3. The paper is well-structured and easy to read.

**Weaknesses:**

1. The core contribution of this paper lies in the CoSplice Module, which further improves temporal consistency. However, the metrics in the Table 1 and 3 reflect a combination of spatial quality and temporal consistency. It is necessary to include comparisons and ablations on metrics that focus specifically on temporal consistency to better demonstrate the contribution of this work.
2. Compared with the first-frame approach, how much does the spliced structure contribute to the performance?  Why do strides of 5 and 20 in Table 3 show no significant performance difference? Do other strides have similar results?
3. The context controller and masked area motion propagation need further analysis and experiments to demonstrate the novelty.

**Questions:**

Please see the weakness.

---

### Note · Authors · 2025-11-12

I have read and agree with the venue's withdrawal policy on behalf of myself and my co-authors.